# Influence of Artificial Aging on Mechanical Properties of Six Resin Composite Blocks for CAD/CAM Application

Wojciech Grzebieluch [1,*], Piotr Kowalewski [2], Mirosław Sopel [3] and Marcin Mikulewicz [4]

1 Laboratory for Digital Dentistry, Department of Conservative Dentistry with Endodontics, Wroclaw Medical University, Krakowska 26, 50-425 Wroclaw, Poland

2 Department of Fundamentals of Machine Design and Mechatronic Systems, Wroclaw University of Science and Technology, Lukasiewicza 7/9, 50-371 Wroclaw, Poland; piotr.kowalewski@pwr.edu.pl

3 Department of Nervous System Diseases, Faculty of Health Sciences, Wroclaw Medical University, Bartla 5, 51-618 Wroclaw, Poland; miroslaw.sopel@umw.edu.pl

4 Department of Dentofacial Orthopeadics and Orthodontics, Division of Facial Abnormalities, Wroclaw Medical University, Krakowska 26, 50-425 Wroclaw, Poland; marcin.mikulewicz@umw.edu.pl

* Correspondence: wojciech.grzebieluch@umw.edu.pl; Tel.: +48-601-725-782

**Abstract:** (1) The interactions in the oral cavity between resin composite blocks for CAD/CAM application and saliva, biofilm, and chemicals and their influence on mechanical properties are still mostly unknown. The purpose of this study is to examine the impact of artificial aging on the flexural strength, flexural modulus, hardness, Weibull modulus, and probability of failure of six resin composite CAD/CAM materials. (2) The aging was conducted by storing the specimens in water at 37 °C for 3 months, then a 3-point bending test was applied and measured. The microhardness was measured with a Vickers microhardness tester. Weibull analysis (according to ISO) was also performed. The shape and scale parameters were calculated. (3) After aging, the flexural strength values ranged from 95.51 (SD 9.07) MPa for the aged Shofu Block HC (HC) to 160.28 (SD 10.37) MPa for non-aged Gandio blocks (GR), and the flexural modulus values ranged from 7.75 (SD 0.19) GPa for HC to 16.77 (SD 0.60) GPa for GR. The microhardness (HV01) ranged from 72.71 (SD 1.43) for the Katana Avencia Block (AV) to 140.50 (SD 5.51) for GR. After aging, the Weibull characteristic strength ranged from 99.47 MPa for HC to 169.25 MPA for Brilliant Crios (CR). (4) Water storage led to a decrease in flexural strength and characteristic strength and slightly affected the flexural modulus. Gandio Blocks, Tetric CAD, and Brilliant Crios presented higher flexural strength than others.

**Keywords:** mechanical properties; dental CAD/CAM; composite resin blocks





## 1. Introduction

The evolution of computer-aided design and computer-aided manufacturing (CAD/CAM) technology has had an impact on dentistry, especially in the field of prosthodontics and restorative dentistry [1]. This development is simultaneous with the advancement of intraoral scanners and clinical protocols, including digital impressions, digital models, virtual articulators, and facebows. The advantages of CAD/CAM technology are based on the simplicity of clinical procedures for indirect dental restoration fabrication at a reduced time and cost [2].

Materials suitable for CAD/CAM applications cover a wide range: titanium, yttrium tetragonal zirconia polycrystals, lithium disilicate glass ceramics, leucite-reinforced glass ceramics, feldspathic glass ceramics, and aluminum oxide [3]. Usually, two classes of materials are used in clinical settings for the production of CAD/CAM restorations, glass-ceramics/ceramics and resin composites. Both materials are delivered in the form of factory-manufactured blocks for subtractive manufacturing. The resin composite blocks for CAD/CAM application are polymerized in factory conditions with high temperature and pressure. Glass-ceramics/ceramics materials present excellent and stable aesthetic and

sufficient mechanical properties, while the resin-based materials have the advantage of also being (after sandblasting) easily repaired in the mouth [4].

The evaluation of mechanical properties and long-term stability in oral conditions in vivo is essential to clarify the clinical potential and indications of new dental materials. However, it is also important to examine these properties in vitro. Conventionally, the mechanical properties of ceramic and resin dental materials are assessed using the basic tests specified by the International Organization for Standardization [5,6].

In vitro tests of new resin composite CAD/CAM materials are needed because clinically validated data are often unavailable because of the time needed to conduct the clinical trials. In many cases, only data published by manufacturers are available. The often-used flexural test method is a 3-point and 4-point bending test, which has a disadvantage due to eliminating unwanted edge failures [7]. Other important characteristics of dental materials include hardness and surface wear [2]. The aging process of these materials has been reported to have a negative impact on their mechanical properties [4,8,9].

It must be underlined that the interactions in the oral cavity between CAD/CAM materials and saliva, biofilm, and chemicals and their influence on mechanical properties are still mostly unknown.

Dental resin composites consist of an inorganic filler and an organic polymer matrix. The connection between these phases is improved by a coupling agent (silanization), and this process reduces water sorption [10,11]. This process, characteristic of resin composites, leads to the degradation of materials and affects their final mechanical properties.

The purpose of this study is to examine the impact of artificial aging on the flexural properties (flexural strength and flexural modulus), microhardness, and Weibull characteristic (modulus and probability of failure) of six resin composites for dental CAD/CAM application.

## 2. Materials and Methods

### 2.1. Study Design

The study was planned in accordance with ISO standards [5,6]. The flexural strength and modulus and microhardness of six different composite block materials for dental CAD/CAM application were tested to analyze the influence of artificial aging. The tested blocks were Grandio Blocs, Tetric CAD, Brilliant Crios, Katana Avencia Block, Cerasmart, and Shofu Block HC. Their composition, according to the literature, is shown in Table 1 [10–15].

**Table 1.** Machinable materials used in the study.

| Brand | Abr. | Manufacturer | Composition | Lot No. | Shade | Block Size |
|---|---|---|---|---|---|---|
| Grandio Blocs | GR | VOCO, Cuxhaven, Germany | 86 wt.% of Nanohybride fillers, 14% UDMA + DMA [10,11] | 1711521 | A2 HT | C 14L |
| Tetric CAD | TE | Ivoclar Vivaden, Schaan, Liechtenstein | Dimethacrylates 28.4 wt.%: of Bis-GMA, Bis-EMA, TEGDMA, UDMA; fillers: 71.1 wt.%, barium glass (<1 μm), silicon dioxide (<20 nm) [12] | 35470 | 3M2 HT | 14 |
| Brilliant Crios | CR | Coltene/Whaledent A.G. Altstatten, Switzerland | Cross-linked resin matrix methacrylate, 70.7 wt.% barium glass (<1 μm), amorphous silica (<20 nm) [10,13] | H22667 | A2 LT | C 14 |
| Katana Avencia Block | AV | Kuray Noritake Dental, Tokyo, Japan | UDMA, TEGDMA with 62 wt.% of aluminum filler (20 nm), silica filler (40 nm) [14] | 000318 | A2LT | 12 |
| Cerasmart | CS | GC Dental Product Europe, Leuven, Belgium | BisMEPP, UDMA, DMA with 71 wt.% of silica (20 nm) and barium glass (300 nm) [10,14,15] | 37690 | A3 C | 14 |
| Shofu Block HC | HC | Shofu Inc., Kyoto, Japan | UDMA, TEGMA, 61 wt.% of silica powder, micro fumed silica, zirconium silicate [10,14,15] | 071601 | A2 LT | 14 |

Bis-EMA: ethoxylate bisphenol A dimethacrylate; Bis-GMA: bisphenol A diglycidylether methacrylate; Bis-MEEP: 2,2-bis(4-methacryloxypolyethoxyphenyl)propane; DMA: dimethacrylate; EDMA–ethyleneglycoldimethacrylate; TEGDMA: triethylene glycol dimethacrylate; UMDA: urethane dimethacrylate.

### 2.2. Sample Fabrication

A Miracut 151 (Metcon, Bursa, Turkey) low-speed water-cooled diamond saw was used to obtain 120 bar-shaped specimens (n = 20). The samples were then finished with a glass grinder (JZO, Jelenia Gora, Poland) with wet silicon carbide (initially 240 ISO/FEPA, average grain size 68 μm, and finally 400 ISO/FEPA, average grain size 35 μm) to obtain 15 mm long, 4 mm wide, and 1.5 mm thick samples with an accuracy of 0.01 mm (according to ISO 6872:2015 5). Due to the limited size of the material blocks, preparation according to ISO 4049 6 was not possible. Half of the samples (n = 10 per material) were tested in dry conditions. The rest were exposed to an artificial aging process (n = 10 per material). The aging was carried out by storing the samples in saline at 37 °C for 3 months. Two groups of preparations were obtained for each material, aged–wet (n = 60) and non-aged–dry (n = 60).

### 2.3. Methods

A three-point bending test was conducted in accordance with ISO 6872:2015 to measure the flexural properties. During the test procedure, the support span was 12 mm, and the loading speed was set to 1 mm/min. A universal testing machine LabTest 5.030S LaborTech® (LaborTech, Opava, Czech Republic), equipped with Test&Motion® (LaborTech Opava, Czech Republic) software, was used (Figure 1) [5].

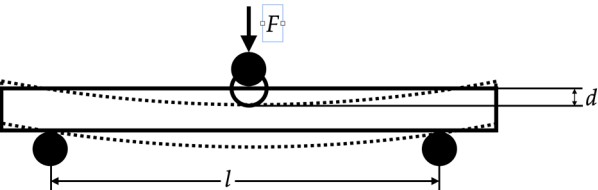

**Figure 1.** Three-point bending test; bar sample and three rollers are visible; (*F*) load; (*l*) roller span; (*d*) deflection (corresponding to load *F*).

Flexural strength ($\sigma_f$) was measured based on the three-point bending tests and was calculated with the formula (in accordance with ISO 6872:2015) [5]:

$$\sigma_f = \frac{3Fl}{2wh^2} \tag{1}$$

where:

*F* is the maximum load recorded during the flexural test, *l* is the span of the roller (12 mm), *w* is the width (4 mm), and *h* is the height (1.5 mm) of the sample bar.

The flexural modulus ($E_f$) was calculated based on the three-point bending test results and using the following formula:

$$E_f = \frac{Fl^3}{4wh^3d} \tag{2}$$

where:

*F* is the load, *l* is the roller span (12 mm), *w* is the width (4 mm), and *h* is the height (2 mm) of the bar, *d* is the deflection corresponding to load *F*.

A Vickers intender tester (Shimadzu HMV-2T, Shimadzu Corp. Kyoto, Japan) was used to measure the microhardness of the materials. During the test procedure, the load was 980.7 mN (HV 0.1), and the dwell time was 10 s. Five indentations were applied at random positions. The surfaces of the samples (the same samples that were used in a three-point bending test) were sequentially polished with the HiLusterPlus® composite rubber polishing system (Kerr Corp., Orange, CA, USA).

### 2.4. Statistical Analysis

Statistical analysis was performed using the Statistica 13 program (StatSoft, Cracow, Poland). For arithmetic variables, arithmetic means and standard deviations were cal-

culated. All quantitative type variables studied were checked with the Shapiro–Wilk test to determine the type of distribution. Comparisons of results between groups (depending on the material used) were carried out using parametric two-factor analysis of variance (ANOVA) together with post hoc testing (Tukey's test). The level $\alpha = 0.05$ was assumed for all comparisons. Weibull statistics were also performed to obtain the shape and scale parameters.

The Weibull modulus ($m$) and the probability of failure ($P_f$) were calculated in accordance with ISO 6872:2015 [5]. The Weibull distribution parameter (Weibull modulus-m) was calculated using the maximum likelihood estimation method. The shape parameter (m, Weibull modulus) and scale parameter (characteristic strength) were calculated. Probability of failure ($P_f$) using the following Equation (3):

$$P_f = 1 - exp\left[-N \cdot \left(\frac{\sigma}{\sigma_0}\right)^m\right] \tag{3}$$

where $m$ is Weibull modulus, $\sigma_0$ characteristic strength, $\sigma$ flexural strength.

This methodology was applied previously [16,17].

## 3. Results

The flexural properties (strength and modulus) and Vickers microhardness of tested materials, aged (wet) and non-aged (dry) are shown in Table 2 and Figures 2–4. A statistically significant difference was found between the means of all parameters studied.

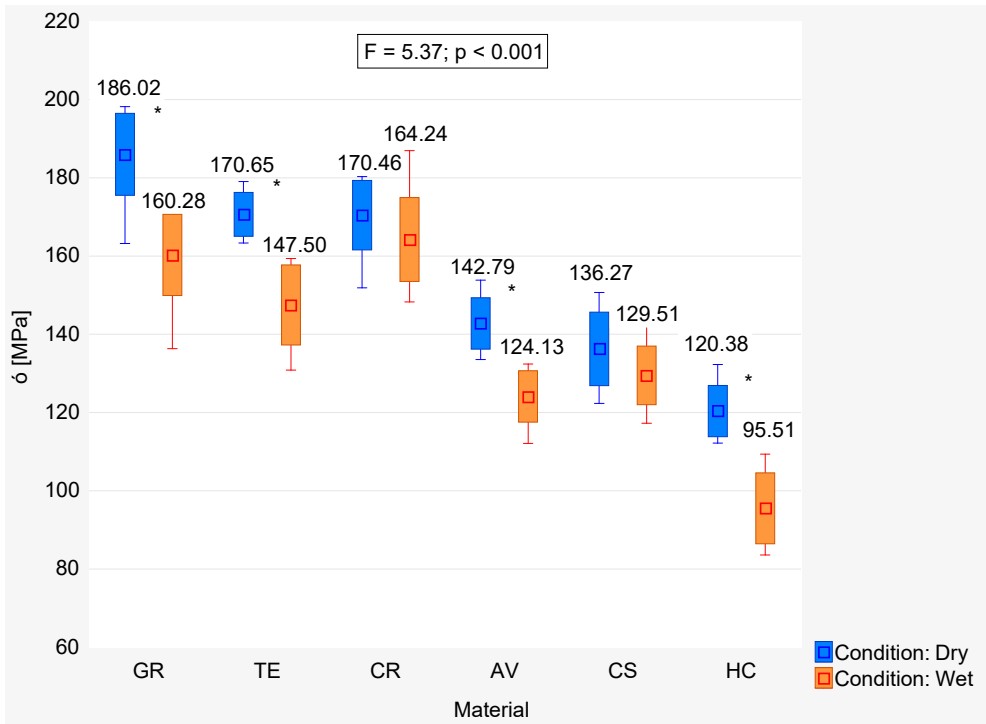

**Figure 2.** Flexural strength of the testing materials before (dry) and after (wet) artificial aging. *—data previously published by the authors.

**Table 2.** Mechanical properties of the testing materials before and after artificial aging.

| Material | | GR | TE | CR | AV | CS | HC | *p*-Value(t) | | |
|---|---|---|---|---|---|---|---|---|---|---|
| **Parameters** | | $\overline{x}$ (SD) | $\overline{x}$ (SD) | $\overline{x}$ (SD) | $\overline{x}$ (SD) | $\overline{x}$ (SD) | $\overline{x}$ (SD) | **Material × Condition** | **Material** | **Condition** |
| $\sigma_f$ [MPa] | Dry | 186.02(10.49) * | 170.65(5.61)A * | 170.46(8.89)AB * | 142.79(6.56)C * | 136.27(9.40)CD * | 120.38(6.54)E * | <0.001 | <0.001 | <0.001 |
| | Wet | 160.28(10.37)ABF | 147.50(10.22)CFG | 164.24(10.72)ABG | 124.13(6.58)DE | 129.51(7.48)DE | 95.51(9.07) | | | |
| | (Δ) | 25.74 | 23.15 | 6.22 | 18.66 | 6.76 | 24.87 | | | |
| | (%) | 13.84 | 13.57 | 3.65 | 13.07 | 4.96 | 20.66 | | | |
| E [GPa] | Dry | 16.95(0.50)A * | 10.56(0.19)B * | 11.14(0.16)C * | 8.39(0.13)D * | 8.45(0.20)D * | 8.26(0.55)D * | 0.001 | <0.001 | 0.006 |
| | Wet | 16.77(0.60)A | 10.84(0.23)BC | 10.96(0.24)B | 7.95(0.09)DE | 8.50(0.13)D | 7.75(0.19)E | | | |
| | (Δ) | 0.18 | −0.28 | 0.18 | 0.44 | −0.05 | 0.51 | | | |
| | (%) | 1.06 | −2.65 | 1.62 | 5.24 | −0.59 | 6.17 | | | |
| HV01 | Dry | 140.43(5.43)A * | 74.88(2.82)BCDE * | 75.40(2.18)BCDE * | 70.85(1.62)E * | 71.13(0.92)BCE * | 77.84(5.11)BCDG * | 0.28 | <0.001 | 0.001 |
| | Wet | 140.50(5.51)A | 75.96(1.49)CDF | 78.16(3.55)D | 72.71(1.43)BCEF | 76.24(2.97)BCDFG | 79.40(3.37)BCDG | | | |
| | (Δ) | −0.07 | −1.08 | −2.76 | −1.86 | −5.11 | −1.56 | | | |
| | (%) | −0.05 | −1.44 | −3.66 | −2.63 | −7.18 | −2.00 | | | |
| *m* | Dry | 25.72 * | 34.89 * | 27.18 * | 22.92 * | 16.35 * | 19.62 * | | - | |
| | Wet | 25.96 | 18.85 | 15.22 | 26.00 | 19.00 | 12.53 | | | |
| | (Δ) | −0.24 | 16.04 | 11.96 | −3.08 | −2.65 | 7.09 | | | |
| | (%) | −0.93 | 45.97 | 44.00 | −13.44 | −16.21 | 36.14 | | | |
| $\sigma_0$ (MPa) | Dry | 190.30 * | 173.26 * | 174.16 * | 145.90 * | 140.50 * | 123.45* | | - | |
| | Wet | 164.22 | 151.91 | 169.25 | 126.89 | 132.94 | 99.47 | | | |
| | (Δ) | 26.08 | 21.35 | 4.91 | 19.01 | 7.56 | 23.98 | | | |
| | (%) | 13.70 | 12.32 | 2.82 | 13.03 | 5.38 | 19.42 | | | |

$\overline{x}$-mean; SD—standard deviation; dry-non-aged; Wet–aged in water; (Δ)—difference (dry/wet) (decrease if positive, increase if negative); (%)—percentage difference wet/dry (decrease if positive, increase if negative); $\sigma_f$—flexural strength, E—flexural modulus; m-shape parameter (Weibull modulus); $\sigma_0$-scale parameter (characteristic strength); Results that are not statistically significant are marked with the same upper case letters (*p* > 0.05; Tukey test); t-Two-way analysis of variance; *—data previously published by the authors [16,17].

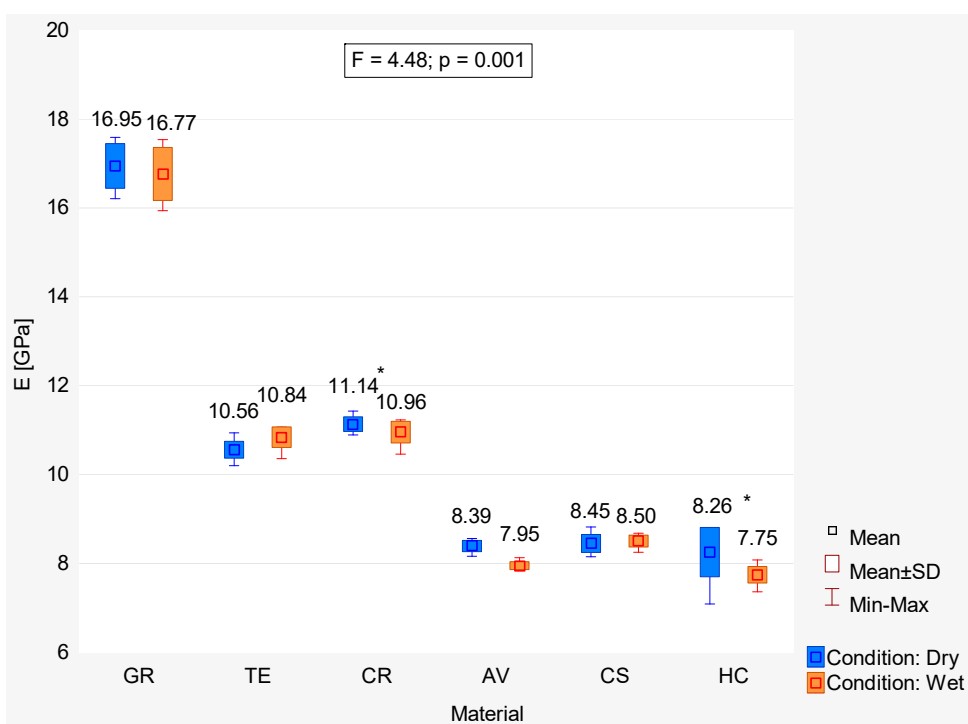

**Figure 3.** Flexural modulus of the testing materials before (dry) and after (wet) artificial aging. *—data previously published by the authors.

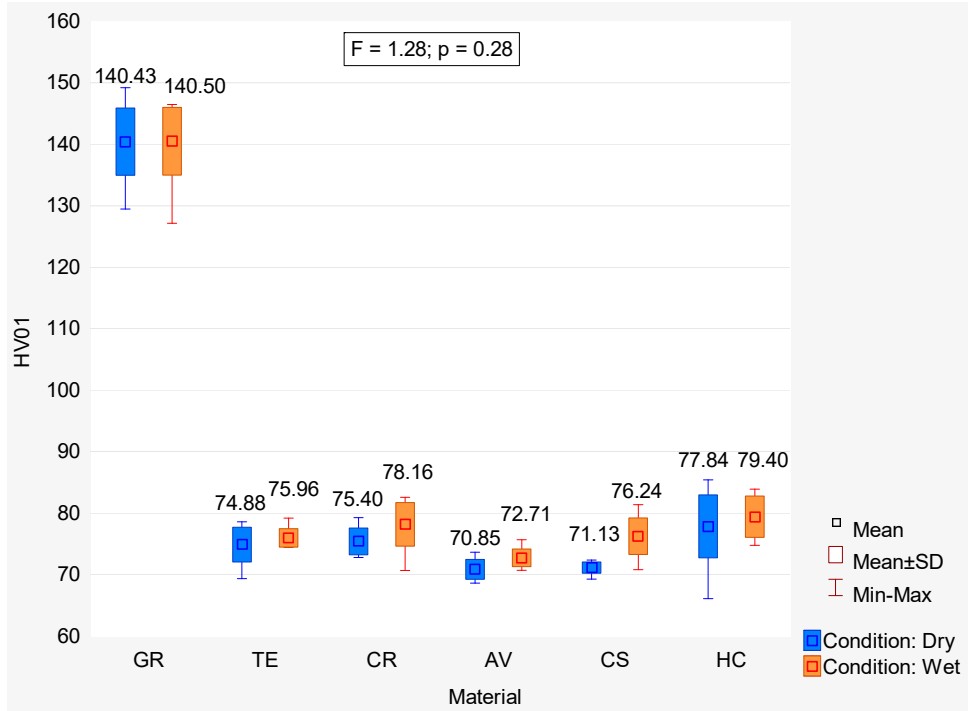

**Figure 4.** Microhardness of the testing materials before (dry) and after (wet) artificial aging.

The values of flexural strength ranged from 95.51 (SD 9.07) MPa for aged HC to 186.02 (SD 10.49) MPa for non-aged GR.

The flexural strength of non-aged GR was significantly higher in comparison to the other tested materials ($p < 0.001$). The aging process caused a decrease in the strength of all tested materials, which ranged from 3.65% for CR to 20.66% for HC. The decrease

was statistically significant for GR, AV, CS, and HC. The aging did not cause a statistically significant decrease in the flexural strength in the case of CR (3.65%) and CS (4.96%).

Flexural strength ranged in decreasing order as follows: GR Dry > TE Dry > CR Dry > CR Wet > GR Wet > TE Wet > AV Dry > CS Dry > CS Wet > AV Wet > HC Dry > HC Wet.

The flexural modulus values ranged from 7.75 (SD 0.19) GPa for HC Wet to 16.95 (SD 0.50) GPa for GR. The recorded GR flexural modulus (Dry and Wet) was significantly higher in comparison to the other tested materials and was also higher for TE (Dry and Wet) and CR (Dry and Wet) compared to AV (Dry and Wet), CS (Dry and Wet), and HC (Dry and Wet). The values of the flexural modulus changed in the descending order as follows: GR Dry > GR Wet > CR Dry > CR Wet > TE Wet > TE Dry > CS Wet > CS Dry > AV Dry > HC Dry > AV Wet > HC Wet.

The aging process caused a decrease in the flexural modulus of GR, CR, AV, and HC, ranging from 1.06% for GR to 6.17% for HC. The decrease was statistically significant for CR and HC. The aging process caused an increase in the flexural modulus of TE (−2.65%) and CS (−0.59%); the changes were not statistically significant.

The microhardness values ranged from 70.85 (SD 1.62) for AV dry to 140.50 (SD 5.51) for GR wet. The microhardness values of GR Wet and GR Dry were significantly higher compared to the other materials tested. The aging process caused an increase in microhardness of all materials tested; the increase in value was from 0.05% for GR to 7.18% for CS. The changes caused by the aging process were not statistically significant. The microhardness values in decreasing order were as follows: GR Wet > GR Dry > HC Wet > CR Wet > HC Dry > CS Wet > TE Wet > > CR Dry > TE Dry > AV Wet > CS Dry > AV Dry.

The calculated Weibull modulus (m) ranged from 15.22 for CR Wet to 34.89 for TE Dry. The ranking of the Weibull modulus was as follows: TE Dry > CR Dry > AV Wet > GR Wet > GR Dry > AV Dry > HC Dry > CS Wet > TE Wet > CS Dry > CRWet > HC Wet. The Weibull survival curves of the flexural strength, showing the probability of failure (Pf) after aging and at any stress level, are shown in Figure 5. Weibull's characteristic strength (σ0) ranged from 99.47 for HC Wet to 190.30 for GR Dry (Table 2, Figure 5). The aging process caused a decrease in the characteristic strength of all materials tested. The values in the diminishing order were as follows: GR Dry > CR Dry > TE Dry > CR Wet > GR Wet > TE Wet > AV Dry > CS Dry > CS Wet > AV Wet > HC Dry > HC Wet.

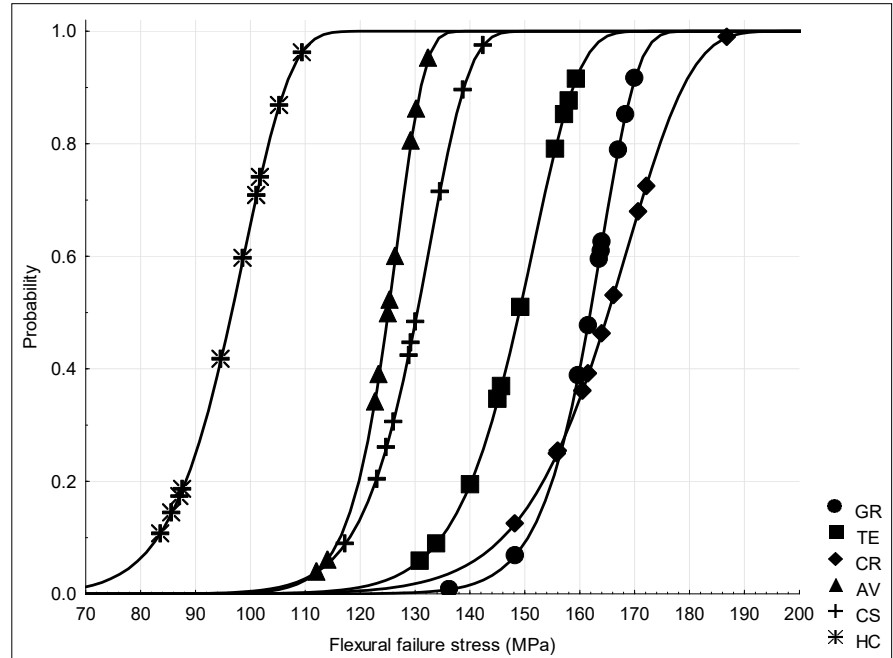

**Figure 5.** Weibull survival curves of the flexural strength of the tested materials after artificial aging (Wet).

## 4. Discussion

In this study, the authors aimed to compare six resin composite blocks designed for chairside CAD/CAM application (resin composites were water-stored for 90 days) and analyze the influence of the aging process on mechanical properties.

Changeable physical–chemical conditions in the oral cavity, such as temperature, pH, biofilm, and dynamic or static loads, create an extremely unfavorable environment for dental materials, which can lead to their chemical degradation. For obvious reasons, it is very hard to examine those materials under in vivo conditions. In in vitro evaluation, it is possible to simulate conditions similar to the oral cavity. In addition to mechanical properties, it is very important to determine the impact of the oral environment on the materials [2].

Some in vitro studies have investigated the chemical degradation of restoratives with food-simulating liquids, and others simulated the aging process with water storage, boiling in water, acid exposure, autoclaving, and thermocycling [2,8]. Water storage has proven to be a simple and effective aging simulation method [8,15].

### 4.1. Flexural Strength

The aging process caused a decrease in the flexural strength of all tested materials. The highest decrease was 20.06% for HC, and the lowest decrease was approximately 13% for GR, TE, and AV. A clearly lower decrease (statistically not significant) by 4.96% and 3.65% in the case of CS and CR, respectively, was recorded. Aged CR, GR, and TE showed significantly higher flexural strength (164.24, 160.28, and 147.50 MPa, respectively) than the others. Egilmez et al. [8] and Lauvahutanon et al. [15] tested the influence of artificial aging on the properties of composite blocks and revealed a high impact of water aging on a significant decrease in the flexural strength of resin composite CAD/CAM materials. The authors reported that the decrease in strength caused by simple water storage aging is comparable to aging by thermocycling. The decrease in flexural strength reported by the authors mentioned above and presented in this study is consistent. However, the strength of CS before aging was similar to those reported by Egilmez et al. [8] and much lower than that registered by Lauvahutanon et al. [15], Awda et al. [18], and Goujat et al. [19]. The strength of the tested materials was lower than reported by Bonner et al. [20] and Niem et al. [21]. The filler properties, volume, and shape of particles affect strength, which allows us to associate the low strength of HC with its structure. According to Okada et al. [22], HC consists of large spherical filler particles that can lead to lower flexural strength. However, an irregular surface of the filler particles disperses more load and also causes the anchor effect, leading to the increased strength of the resin composites.

### 4.2. Flexural Modulus

The stress–strain characteristic of the restorative material is an important factor. Flexural properties related to the stress–strain characteristics of the tooth-restoration complex and mismatch in this area can lead to discontinuity in load transfer and can be the reason for failures. For this reason, the measurement and analysis of the E modulus are crucial. The abovementioned parameter is represented by the slope of the part of the stress–strain curve recorded during the 3-point bending test and describes the stiffness [5,18]. In the case of indirect restorations, the E modulus is a very important parameter because of its crucial impact on the occlusal load distribution in the hard tissues and the tissue-restoration interface. Composite and ceramic materials differ significantly in stiffness despite the fact that the indications for their use often overlap [18,19]. Ceramic materials with high E-modulus values transfer less stress to the remaining structures and are potentially less prone to debonding than resin composites [23]. On the other hand, CAD/CAM resin composites show better machinability and lower marginal edge roughness than stiffer ceramic materials [18]. According to Masouras et al. [24], the flexural modulus of resin composites is correlated with filer volume ($r^2 = 0.9$), so the higher the filer content (by vol), the higher the flexural modulus values. GR was found to be the stiffest among the tested materials,

and the flexural modulus of this material was more similar to that of dentin, according to Kinney et al. [25]. All materials tested, in terms of stress–strain characteristics, differed significantly from enamel [26–28]. The results obtained showed low changes in flexural modulus after the artificial aging process. A slight but statistically significant impact was observed only on CR and HC. The modulus values obtained were consistent with those reported in the research papers [10,11,13,15,18,20,21]. The exception is the results presented by Goujat et al. [19], where the CS flexural modulus reaches 25 GPa, even higher than reported for polymer infiltrated ceramic network (PICN) Enamic (Vita, Bad Sackingen, Germany). PICN is a new type of resin-containing material; its filler is a porous and solid ceramic structure infiltrated by the resin. Its unique construction makes this material much stiffer compared to the tested powder/resin CAD/CAM composite materials, with an E modulus of over 20 GPa [10,15,18].

*4.3. Microhardness*

Of the tested materials, the GR showed the highest value of hardness, exceeding 140. This material reaches a value almost two times higher than those of other tested blocks. The values recorded for other materials did not differ significantly. It is also worth mentioning that the water aging process caused only slight and not statistically significant changes in the microhardness of all resin composites tested. Alamoush et al. [10] also did not record a significant difference between the microhardness of CR, CS, and HC, and they reported that the GR was the hardest. However, the difference in microhardness between GR and the other composites was smaller than reported in Table 2. The results obtained in the present study are also consistent with the data by Lauvahutanon et al. [15], who also found no significant difference in hardness between CS and HC. Returning to the properties of GR, its highest stiffness and hardness can be associated with very high filler content (84.6–86% by wt.) and densely packed, irregular, nonrounded filler grains of varying sizes [10,11].

The lower hardness value of resin composite blocks compared to ceramic materials contributes to their better milling ability [4].

*4.4. Weibull Statistic*

Weibull statistical analysis is used to examine variations in strength and the homogeneity of materials. It is based on the Weibull modulus (*m*) value, which is related to the flaw density and sizes. A higher m value means smaller deviations of flexural strength as a result of a lower density of internal flaws such as air bubbles and cracks. Dental materials with a higher m value are preferable for this reason, even if a higher (*m*) value is associated with slightly lower strength. Another advantage of the Weibull statistical approach is the determination of the failure probability at any possible stress level. This analysis makes it possible to evaluate the reliability of materials and evaluate reliability as a function of the load. The Weibull characteristic strength value allows calculating strength value with the failure probability equal to 63.2% ($P_f$ = 63.2%) [7,29–33]. The *m* values of the CAD/CAM resin composites recorded in this study and reported by Lim et al. [30] were higher than the reported m value obtained by Bonna et al. [29] and Rodtiguez Junior et al. [32] for light-cured micro-hybrid and nanofill composites. This may suggest better homogeneity and lower flaw density of factory polymerized resin composite materials. Water storage decreases the m values of TE, CR, and HC by 45.97, 44.00, and 36.14%, respectively. At the same time, the Weibull modulus of GR, AV, and CS increases by 0.9%, 13.44%, and 16.21%, respectively (Table 2). Water aging had a similarly negative effect on flexural strength and caused a decrease in the Weibull characteristic strength of all tested materials (Table 2, Figure 5).

*4.5. Influence of Water Storage on Flexural Properties*

All contemporary resin composites are characterized by hydrophilicity and hydrolytic effect [33]. Water storage affects the flexural properties of composite materials, slowly penetrating into the structure of the resin composite. Longer storage times in water may

cause greater changes in properties [34,35]. The rate and amount of water absorption depend on the resin composition [33].

The smallest changes in flexural properties among the tested materials were shown by CS. This can be explained by the different compositions of the resin containing BisMEPP and NPG. CR, with the lowest decrease in flexural strength after water storage, contains bis-GMA as TE, Bis-EMA as TE, and TEGDMA as TE and AV. The highest changes of flexural properties among the tested materials were shown by HC, wherein the resin consists of a mixture of UDMA and TEGDMA. The UDMA monomer is also included in GR, TE, and AV, and the highest flexural strength changes were observed for materials containing this monomer.

## 5. Conclusions

Taking into account the limitations of the present in vitro study, it can be concluded that artificial aging in water:

- Led to a statistically significant decrease in flexural strength of GR, TE, AV, and HC;
- Caused no significant decrease in CR and CS flexural strength, and GR, TE, and CR presented higher flexural strength than others;
- Caused a slight increase in the microhardness of tested resin composite blocks;
- Had a low impact on flexural modulus;
- Caused a decrease in the Weibull characteristic strength, i.e., an increase in the probability of failure.

**Author Contributions:** Conceptualization, W.G.; methodology, W.G. and P.K.; validation, W.G., P.K. and M.M.; formal analysis, M.M. and M.S.; investigation, W.G., P.K. and M.S.; resources, W.G.; data curation, M.S.; writing—original draft preparation, W.G., P.K., M.S. and M.M.; writing—review and editing, W.G. and M.M.; visualization, W.G., P.K. and M.S.; supervision, M.M.; project administration, W.G.; funding acquisition, W.G. All authors have read and agreed to the published version of the manuscript.

**Funding:** The research was supported by the Medical University of Wroclaw, grant ST.B010.17.02.

**Institutional Review Board Statement:** Not applicable.

**Informed Consent Statement:** Not applicable.

**Data Availability Statement:** Data available from the authors: wojciech.grzebieluch@umw.edu.pl.

**Conflicts of Interest:** The authors declare no conflict of interest.

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
