# Peer review of "Influence of Artificial Aging on Mechanical Properties of Six Resin Composite Blocks for CAD/CAM Application"

_coatings, doi:10.3390/coatings12060837_

Round 1

Reviewer 1 Report

  1. English should be improved. Some sentences are hard to understand. Please pay attention to writing, for examples: “properties in in vitro condition”; table caption of Table 2 “Wet - aged; wet – aged;”; “The values of in the diminishing order”; In in vitro valuation,”; “The clearly stated lower decrease in flexural strength”; “the more filer by vol”; “between microhardness CR, CS and HC”; “higher than reported”
  2. The title seems inappropriate, and it is suggested to change “CAD/CAM”.
  3. “CAD/CAM materials” in Abstract should be explained first.
  4. “CAD/CAM materials” should be defined explicitly in Introduction section.
  5. It is better to illustrate what deflection d is in Eq. (2) by a schematic drawing.
  6. More sentences are needed to explain Eq. (3). Why is this type of Weibull equation used? What are the physical background to the fitting parameters?
  7. The symbols in table caption of Table 2 are not correct.
  8. Please pay attention to physical symbols, many of them are not represented in the correct way.
  9. More science is needed, and the experimental results should be explained in a scientific way.

Author Response

Dear Reviewer,

We would like to thank You for Your valuable suggestions which improved our study. We have performed following changes:

  1. English should be improved. Some sentences are hard to understand. Please pay attention to writing, for examples: “properties in in vitro condition”; table caption of Table 2 “Wet - aged; wet – aged;”; “The values of in the diminishing order”; In in vitro valuation,”; “The clearly stated lower decrease in flexural strength”; “the more filer by vol”; “between microhardness CR, CS and HC”; “higher than reported”

 Response
“properties in in vitro condition” has been changed to “properties in vitro”

 “Wet - aged; wet – aged;” has been changed to “Wet – aged in water”

“The values of in the diminishing order” has been changed to “The values in the diminishing order”

“The clearly stated lower decrease in flexural strength” the sentence has been changed to:

“The clearly stated lower decrease (statistically not significant) by 4.96% and 3.65% in the case of CS and CR, respectively was recorded.”

“the more filer by vol the higher flexural modulus values” the sentence has been changed to:

“According to Masouras et al. [24] flexural modulus of resin composites is corelated with filer volume (r2 = 0.9), so the higher the filer content (by vol) the higher flexural modulus values.”

“between microhardness CR, CS and HC” – the sentence has been rewritten : “Alamoush et al. [10] also did not record a significant difference between microhardness of CR, CS and HC and they reported that the GR was the hardest.”

“higher than reported” – the sentence has been rewritten: ‘The m values of the CAD/CAM resin composites recorded in this study and reported by Lim et al. [31] were higher than reported than m value obtained by Bonna et al. [30] and Rodtiguez Junior et al. [35] for light-cured microhybrid and nanofill composites. This may suggest better homogeneity and lower flaw density of factory polymerized resin composite materials.”

  1. The title seems inappropriate, and it is suggested to change “CAD/CAM”.

 Response
Title has been changed to: “Influence of Artificial Aging on Mechanical Properties of Res-in Composite Blocks for CAD/CAM application.“

  1. “CAD/CAM materials” in Abstract should be explained first.

 Response

The “CAD/CAM materials” has been replaced with “resin composite blocks for CAD/CAM application” to explain that the materials were delivered in the form of factory polymerized blocks.

  1. “CAD/CAM materials” should be defined explicitly in Introduction section.

 Response

Introduction has been improved. “CAD/CAM materials” has been explained by adding: “Both materials are deliered in the form of factory manufactured blocks for subtractive manufacturing. The resin composite blocks for CAD/CAM application are polymerized in factory conditions by high temperature and pressure.”

  1. It is better to illustrate what deflection d is in Eq. (2) by a schematic drawing.

 Response
The schematic drawing has been added.

  1. More sentences are needed to explain Eq. (3). Why is this type of Weibull equation used? What are the physical background to the fitting parameters?

 Response
Weibull analysis was conducted according ISO ….. Application of Eq. (3) results, in our opinion, in more clear and easy to understand by desnists graph (Fig. 4 and 5 – afrer corrections).

  1. The symbols in table caption of Table 2 are not.  correct.

 Response
Tabele 2 Has been revised and improved.

  1. Please pay attention to physical symbols, many of them are not represented in the correct way.

Response

The physical symbols has been revised and improved. The font format has been revised.

  1. More science is needed, and the experimental results should be explained in a scientific way.

 Response
Discussion has been extended and conclussions has been improved.

Best regards,

Authors.

Reviewer 2 Report

Authors present an interesting paper in which they examine the influence of aging process on mechanical properties of  6 resin composite CAD/CAM materials. The manuscript is well documented with important results.

I would recommend a graph showing differences between the results obtained for flexural strength and microhardness in both wet and dry samples. But this remark is at the discretion of the authors. 

Conclusions section should be improved and provide a better overview of the study. 

Minor modifications should be considered by authors - e.g.:

Line 129 - “Wet” is specified two times. 

Line 138 - Description should be completed: “…(wet) artificial” aging.

Author Response

Dear Reviewer,

We would like to thank You for Your valuable suggestions which improved our study. We have performed following changes:

Authors present an interesting paper in which they examine the influence of aging process on mechanical properties of  6 resin composite CAD/CAM materials. The manuscript is well documented with important results.

I would recommend a graph showing differences between the results obtained for flexural strength and microhardness in both wet and dry samples. But this remark is at the discretion of the authors. 

Response

Thank you for your suggestions, we decided to leave Figs 2-4 unchanged.

Conclusions section should be improved and provide a better overview of the study. 

Response

Conclusions has been revised and extended.

Minor modifications should be considered by authors - e.g.:

Line 129 - “Wet” is specified two times. 

Response

Has been corrected.

Line 138 - Description should be completed: “…(wet) artificial” aging.

Response

Has been corrected.

Reviewer 3 Report

This paper basically deals with the measurement of properties of some well-known dental materials that the authors procured from manufacturers. They aged the samples by immersing in water for 3 months at 37C which is not even close to similar to the oral cavity and especially compared to the average lifetime of dental implants which is 20 years. Even if they consider just immersing in a liquid for 3 months, it should be saline and not just water. Thus, the aging conditions are not at all favorable for inducing aging on the materials - which is the biggest drawback of the manuscript. The authors should consider autoclaving for 4h to mimic aging on the implants. 

Also, the discussion of the results is not good. The authors do not compare the property changes after it has been immersed in water. 

Author Response

Dear Reviewer,

We would like to thank You for Your valuable suggestions which improved our study. We have performed following changes:

This paper basically deals with the measurement of properties of some well-known dental materials that the authors procured from manufacturers. They aged the samples by immersing in water for 3 months at 37C which is not even close to similar to the oral cavity and especially compared to the average lifetime of dental implants which is 20 years. Even if they consider just immersing in a liquid for 3 months, it should be saline and not just water. Thus, the aging conditions are not at all favorable for inducing aging on the materials - which is the biggest drawback of the manuscript. The authors should consider autoclaving for 4h to mimic aging on the implants. 

Response

Thank you for evaluating our work. We agree that immersion in water does not fully reflect oral conditions. However, for many reasons, both technical and economic, some simplification is necessary. Various aging methods are found in publications. There are also publications that compare different aging methods. We made the choice based on results of Egilmez et all. Study. These authors analyzed the effect of different aging methods on composite blocks 3. Water immersion is comparable with thermal cycling and boiling in water. It has also higher impact on flexural properties than acid storage. Therefore, we concluded that soaking in water would effectively and easily capture changes in material properties. Since all materials were tested under simmilar conditions, it allows for the detection of differences (visible in the results) and drawing conclusions. As users of the assessed materials, we believe that this simple study also has practical implications. Especially when new materials appear on the market, or are modiffied modified (such as Cerasmat). 

EGILMEZ, Ferhan, et al. Does artificial aging affect mechanical properties of CAD/CAM composite materials. Journal of prosthodontic research, 2018, 62.1: 65-74.

Also, the discussion of the results is not good. The authors do not compare the property changes after it has been immersed in water. 

Response

Discussion has been revised and extended; conclussions has been improved.

Round 2

Reviewer 3 Report

The authors have addressed my comments. Minor editing needs to be done for the English language.